**Subject Category:**
Biology (whole organism)

ecology/health and disease and epidemiology

biological invasion, emerging infectious disease, One Health

**Author for correspondence:**
Nick H. Ogden
e-mail: nicholas.ogden@canada.ca

# Emerging infectious diseases and biological invasions: a call for a One Health collaboration in science and management

Nick H. Ogden[1,2], John R. U. Wilson[3,4], David M. Richardson[3], Cang Hui[5,6], Sarah J. Davies[3], Sabrina Kumschick[3,4], Johannes J. Le Roux[3,7], John Measey[3], Wolf-Christian Saul[3,5] and Juliet R. C. Pulliam[2]

[1]National Microbiology Laboratory, Public Health Agency of Canada, Canada
[2]South African DST-NRF Centre of Excellence in Epidemiological Modelling and Analysis (SACEMA), Stellenbosch University, South Africa
[3]Centre for Invasion Biology, Department of Botany and Zoology, Stellenbosch University, South Africa
[4]South African National Biodiversity Institute, Kirstenbosch Research Centre, Claremont, Cape Town, South Africa
[5]Centre for Invasion Biology, Department of Mathematical Sciences, Stellenbosch University, Matieland 7602, South Africa
[6]Mathematical and Physical Biosciences, African Institute for Mathematical Sciences (AIMS), Muizenberg 7945, South Africa
[7]Department of Biological Sciences, Macquarie University, Sydney 2109, Australia

NHO, 0000-0002-1062-7283; JRUW, 0000-0003-0174-3239;
DMR, 0000-0001-9574-8297; CH, 0000-0002-3660-8160;
SK, 0000-0001-8034-5831; JJL, 0000-0002-9529-6451;
JM, 0000-0001-9939-7615; W-CS, 0000-0002-3584-6159;
JRCP, 0000-0003-3314-8223

The study and management of emerging infectious diseases (EIDs) and of biological invasions both address the ecology of human-associated biological phenomena in a rapidly changing world. However, the two fields work mostly in parallel rather than in concert. This review explores how the general phenomenon of an organism rapidly increasing in range or abundance is caused, highlights the similarities and differences between research on EIDs and invasions, and discusses shared management insights and approaches. EIDs can arise by: (i) crossing geographical barriers due to human-mediated dispersal, (ii) crossing compatibility barriers due to evolution, and (iii) lifting of environmental barriers due to environmental change. All these processes can be implicated in biological invasions, but only the first defines them. Research on EIDs is

embedded within the One Health concept—the notion that human, animal and ecosystem health are interrelated and that holistic approaches encompassing all three components are needed to respond to threats to human well-being. We argue that for sustainable development, biological invasions should be explicitly considered within One Health. Management goals for the fields are the same, and direct collaborations between invasion scientists, disease ecologists and epidemiologists on modelling, risk assessment, monitoring and management would be mutually beneficial.

# 1. Background

Changes to climate, habitats and biodiversity are affecting abiotic and biotic components of ecological niches, while social and economic changes (e.g. the development of megacities and increasing movement of people and goods in a globalized world) offer multiple routes for species translocation and dissemination [1–3]. Together these external drivers increasingly facilitate biological invasions, a major threat to biodiversity and ecosystems globally [4]. Non-native species include disease-causing microorganisms and parasites, and disease vectors (e.g. arthropod vectors such as mosquitoes), which pose substantial threats to human, domesticated animal and wildlife populations. Invasions by pathogens are, in public and animal health terms, emerging infectious diseases (EIDs; such as human immunodeficiency virus (HIV) and severe acute respiratory syndrome (SARS)) [5,6]. In this paper, we focus on the mutual relevance of invasion science [7] and public health epidemiology in the context of EIDs of direct public health significance [8]. We also highlight how invasive non-pathogenic species, and infectious diseases that do not affect humans or domesticated animals directly, may indirectly impact human health. Possible indirect effects include those affecting the health of domesticated animals, crops, natural resources of wild plant and animal origin and also the health of natural ecosystems. Epidemiology is a broad field that encompasses many areas of health research; here, we use the term 'epidemiologists' to refer to those within the subspecialty focused on epidemiology of EIDs, which may also include disease ecologists. Responses to EIDs engage a wide community of medical, veterinary and public and animal health professionals.

The World Health Organization (WHO) defines an EID as 'an infectious disease that has appeared in a population for the first time, or that may have existed previously but is rapidly increasing in incidence or geographic range' (https://apps.who.int/iris/handle/10665/204722). Infectious diseases emerge via a number of mechanisms. 'Adaptive emergence' constitutes genetic change of a microorganism that results in a phenotype that is capable of invading a new ecosystem, particularly by jumping to new host species, including humans [9]. This mechanism of emergence may permit pathogens causing animal infections to become transmissible to humans (i.e. become zoonoses) and, in some cases, to be sustained by human-to-human transmission in the absence of animal reservoir hosts [10–12]. Expansion or 'geographical emergence' by changes to geographical ranges of pathogens or parasites can involve long-distance translocation, more localized spread or both. For invasion biologists, invasive species are those translocated intentionally or accidentally through a human agency (often over long distances) from the locations where they are native to an ecosystem where they were previously absent [13,14]. This is analogous to the emergence of EIDs by long-distance geographical spread.

The ideas that EIDs are essentially invasive species [15], and that two branches of science (invasion science and EID epidemiology) are studying similar phenomena [16–18], are not new. Furthermore, management objectives and methods may be similar [19]. Invasive arthropod vectors of parasites and pathogens, such as *Aedes* species of mosquitoes, are a case in point; they are traditionally considered part of EID studies, but are also studied by invasion biologists (e.g. [17,20]). However, despite these commonalities, functionally, the fields of invasion science and EID epidemiology work in parallel rather than together. Therefore, in this review, we explore the extent of similarities in key concepts, processes and methodological approaches, as well as useful differences that provide opportunities for synergies, which may enhance our understanding and practical management of invasions and EIDs. We call for these fields to be integrated within the One Health approach to enhance human well-being.

# 2. Common ground

## 2.1. Shared global context: the One Health concept

EIDs that have affected humanity in recent decades have sharpened the focus of microbiologists, epidemiologists, human and animal health practitioners, as well as environmental and biological

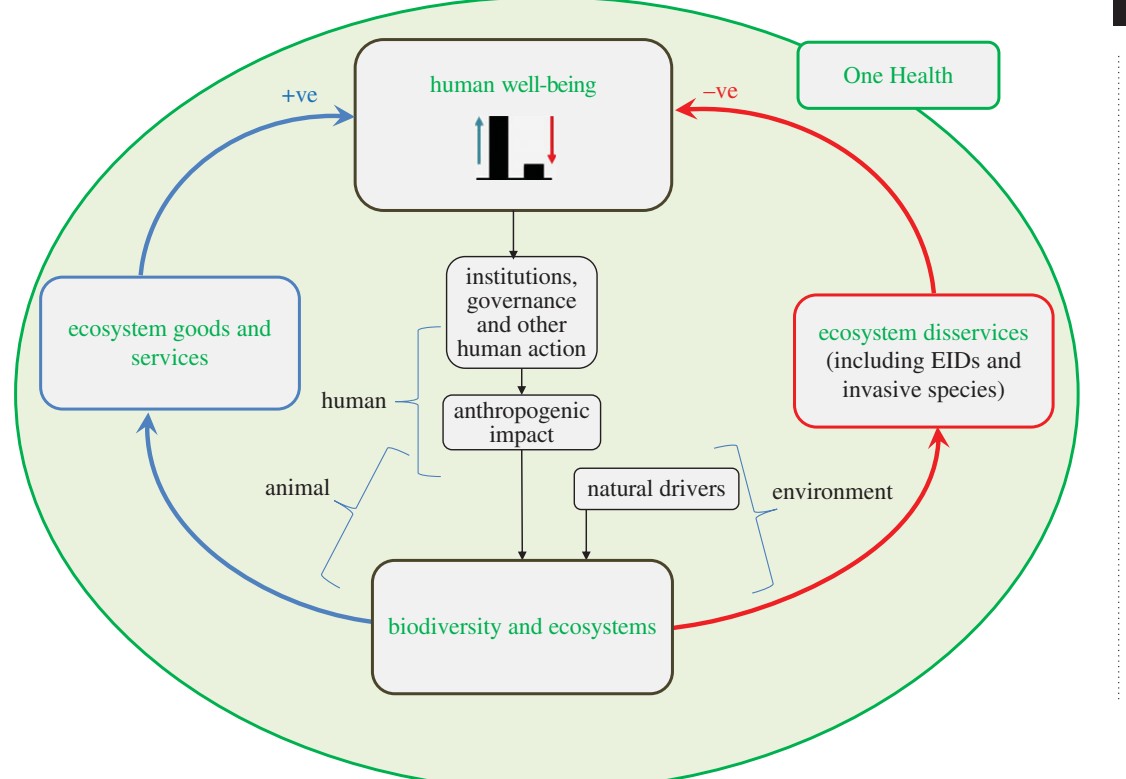

**Figure 1.** Biological invasions and EIDs as components of One Health. The schematic combines an adaptation of the IPBES Conceptual Framework [29] with a schematic of the One Health concept. The IPBES Conceptual Framework illustrates the interplay between anthropogenic and natural drivers of change in nature (biodiversity and ecosystems) (black boxes and arrows) and how this connects ecosystem services to human well-being (+ve effects, blue box and arrows). We also identify connections to ecosystem disservices, such as those caused by EIDs and invasive species (−ve effects, red box and arrows). For simplicity, positive effects of invasive species are not shown. The One Health concept (green circle) encompasses the IPBES Conceptual Framework, with its interacting human, animal and environment components.

scientists, on the intersections of human, animal and ecosystem health. Emergence of many infectious diseases is associated with the dynamics of natural communities and their abiotic environmental determinants [21]. Many EIDs, including invasive pathogens such as West Nile virus (WNV) in North America, are maintained by (or originate in) wild animal hosts, and their emergence may have negative effects on natural communities as well as human or production animal health [22]. Accordingly, the One Health concept has evolved, which postulates that human, animal and ecosystem health are interrelated and interdependent, and that reactionary or preparatory responses to threats to human well-being demand holistic, transdisciplinary approaches encompassing all three components, including medical and veterinary practitioners and collaborators in ecosystem health [23]. Public health organizations around the world are increasingly adopting the One Health approach to make their responses to infectious diseases more effective (e.g. https://www.cdc.gov/onehealth/). The One Health concept encompasses benefits to human well-being (ecosystem services, i.e. benefits produced by ecosystem functions and structures for human well-being) as well as risks (ecosystem disservices, i.e. 'nuisances' for human well-being such as pests, and biological and geophysical hazards [24]). Both EIDs [25] and biological invasions [26] are important causes of ecosystem disservices, although biological invasions often render services and disservices at the same time [26]. Both disease emergence and biological invasions are increasing, being driven by the same global changes in climate, biodiversity, socio-economics and trade/travel [27]. The Intergovernmental Platform on Biodiversity and Ecosystem Services (IPBES, www.ipbes.net) was launched in 2012 to assess the state of biodiversity and of the ecosystem services it provides to society [28,29]. Integrating disservices in the IPBES conceptual framework illustrates the shared role that EIDs and biological invasions play for human well-being as components of One Health (figure 1).

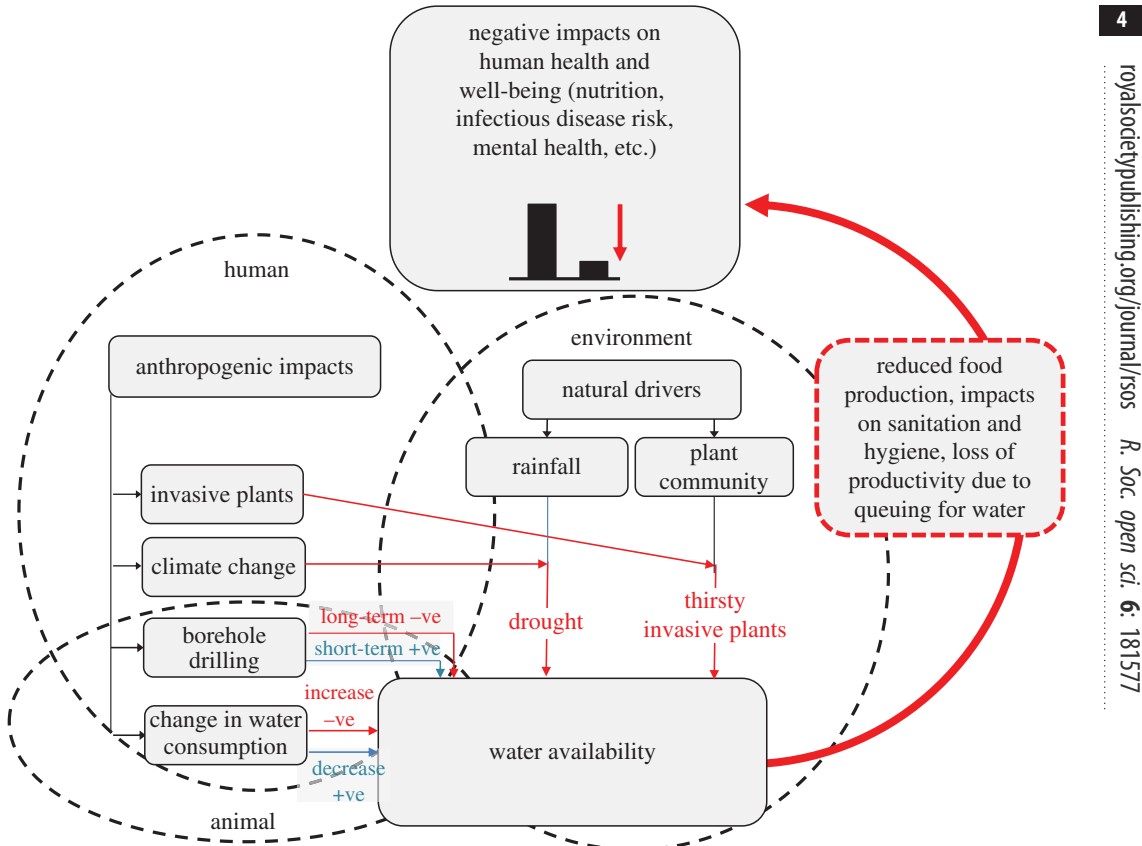

**Figure 2.** An example of indirect effects of invasive species on human health. Here, the indirect impact is water availability, which in South Africa is imperilled by invasive plants that are 'thirsty' (i.e. take up water at rates that significantly reduce water flows), climate change-induced drought and the competing requirements of drinking water for human populations, livestock production and other agricultural enterprises. How this issue is central to the One Health concept is illustrated by the interacting human, animal and environment components of the water availability problem as indicated by dashed circles. These circles indicate the main impacts of humans (the anthropogenic impacts), animals (the consumption of water by livestock and the consequent need to drill boreholes) and the environment (rainfall and plant communities).

The One Health concept recognizes that impacts of ecosystem changes on human health may act indirectly, e.g. via impacts on food and water security or by affecting biodiversity [17,30,31], and the UN Food and Agriculture Organization has adopted the One Health approach (http://www.fao.org/asiapacific/perspectives/one-health/en/). For example, several invasive trees in South Africa reduce water availability, thereby causing indirect impacts on human health (figure 2). More generally, biological invasions are increasingly being framed in a context of a transdisciplinary social–ecological system in which wider implications, including health and socio-economic impacts, are considered [32]. In South Africa, such transdisciplinary approaches have been termed 'invasion science for society' [33], which echoes the One Health concept.

## 2.2. Common drivers and biological processes

There are many overlaps and parallels between EIDs and biological invasions. Both involve species crossing geographical barriers that historically prevented natural dispersal, processes of establishment in a new environment, and subsequent range expansion to occupy the new environment. Not all EIDs can be termed invasive species, but some EIDs spread, and many establish, internationally, and such pathogens can be readily considered as invasive species (e.g. WNV, chikungunya, SARS and Zika in the Americas; chikungunya and dengue in Europe; HIV and influenzas globally). Even if EID emergence is associated with native range expansions (e.g. the spread of Lyme disease into Canada from the USA), and as such might not be formally considered as invasive species, insights on the basis of invasion concepts are still very relevant.

**Table 1.** Barriers to invasions and disease emergence, the processes whereby these may be surmounted and the phenomena and consequences that may result. EID, emerging infectious diseases.

| initial barrier which when crossed can lead to the phenomenon | process | global change examples/mechanisms | EID examples |
|---|---|---|---|
| geographical | dispersal | biological invasions (i.e. inter-regional dispersal of alien species by humans) | EIDs involving international spread (e.g. HIV, SARS, WNV) |
| compatibility | evolution | pre-adaptation via eco-evolutionary experience. Evolution of new phenotypes in the environment (e.g. herbicide resistance, reduction in body size due to size-selected harvesting, new associations) | adaptive emergence of a zoonosis (e.g. zoonotic influenza) greater capacity to survive and reproduce, allowing species to spread (e.g. WNV in North America) |
| environmental | disturbance | land-use change that removes competitors or predators, or opens up resources allowing range expansion of species (native or non-native) | provides new opportunities for contact between humans, animals and disease vectors; and causes biodiversity change driving disease emergence |
| | | climate change that changes the geographical location of the ecological niche of species | diseases and their vectors (e.g. Lyme disease vectors in Canada) |

The concepts of 'barriers' and 'stages' are as relevant in biological invasions [13] as they are to disease emergence by international spread of a pathogen [5], and also to the processes mediating de novo emergence of a zoonosis from a microorganism maintained by animal reservoir hosts [34]. This topic has been reviewed before [17,35]. However, we focus on three key elements that permit, or prevent, EIDs and biological invasions: (i) geography, which is surmounted by dispersal; (ii) compatibility, which is determined by genetics and may be surmounted by evolution (including pre-adaptation via eco-evolutionary experience; see below); and (iii) environment, which is a barrier that may be lifted by disturbance, including environmental changes. Together, these factors mediate the biotic and abiotic qualities of the niche, the species' fitness in that niche and determine how the niche qualities and fitness may change (table 1 and figure 3).

(i) **Geography**: The crossing of historical geographical barriers and human-mediated introductions are related to both invasive species and many EIDs. The movement of invasive species, and long-distance dispersal of EIDs or their vectors, occurs via air and surface transport of goods and people [36]. For infectious diseases of humans, air travel is considered the most important route because it is rapid enough for humans infected in source locations to remain infective upon arrival in their destinations (e.g. SARS [37]). For many invasive species, the travel time from the native to the alien region is less important due to the occurrence of long-lived life stages such as seeds and eggs, so international spread of plants and animals is often facilitated by surface transport (on land or by sea). However, surface transport is also important for EIDs whereby infected arthropods, invasive arthropod vector eggs and infected animal hosts may be transported over long distances, e.g. the historical spread of plague and the recent spread of *Aedes albopictus* eggs/immatures in tyres and house plants [38–40]. While not a typical feature of EID introductions, deliberate transport and introduction of invasive species is common [41,42]. Also, both EIDs and invasive species have a history of, and the potential for, being introduced

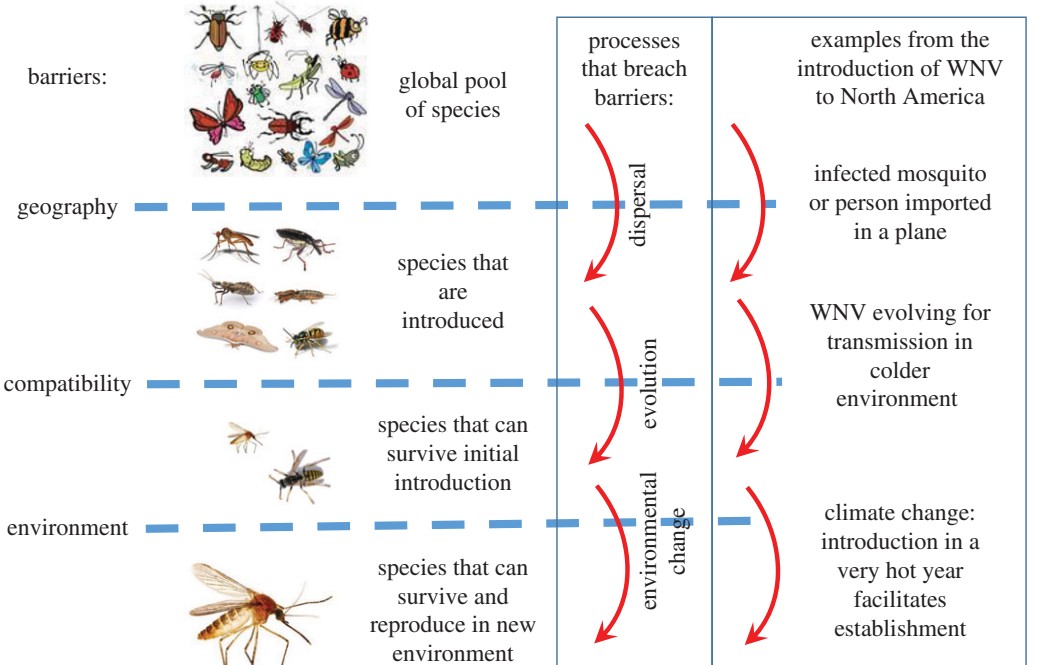

**Figure 3.** A conceptual diagram of the barriers to biological invasions and EIDs and how they limit species invasions and disease emergence. Processes whereby barriers may be breached are shown in the central box, and an example of these (from the introduction of West Nile virus (WNV) to North America) is shown in the box to the right. Note the only prerequisite for biological invasions is that there is dispersal across a geographical barrier (evolution and environmental change are not required if conditions are already suitable). By contrast, an EID can arise either through evolution leading to the breakdown in a compatibility barrier or environmental change breaking an environmental barrier without there being dispersal over a geographical barrier (cf. table 1). Moreover, the order of the barriers crossed can vary. For example, in the emergence of HIV, a compatibility barrier was first crossed (non-human primate to human) before the global spread of the pandemic. The insect collage used under 'species that are introduced' in Figure 3 was sourced from Wikimedia Commons under the Creative Commons Share-Alike License (CC-BY-SA 3.0; see https://commons.wikimedia.org/wiki/File:Insect_collage.png). We acknowledge the original author of the work: 'BugBoy52.40'.

via the international pet trade [43,44], and both may be introduced deliberately as acts of bioterrorism (e.g. [45]). The bridging of the 'geographical' contact barrier between animals and humans (a process known as 'spillover') is essential for the de novo emergence of microorganisms as zoonoses, and the re-emergence of many zoonoses such as the spread of Nipah and Hendra viruses to humans (who are readily infected by the virus) from wildlife reservoirs [18]. Many zoonoses and arthropod vectors are dispersed regionally or more locally by natural means, which are not usually considered in the context of invasive species. Dispersal by migratory birds is one important mechanism whereby pathogens (e.g. influenza viruses) and some disease vectors (particularly ticks) can be dispersed over long distances (e.g. [46]).

Beyond the simple contingency of species being transported into a new environment, the number and size of introduction events of a given species is also important. This is termed propagule pressure in invasion science and is analogous to concepts of infection frequency (relevant for spillover and introduction to new areas) and infective dose that are important in infectious disease epidemiology [13,17,47]. If propagule pressure is low, introduced species are more likely to undergo stochastic fade-out for a range of reasons, including the probability that an infected individual meets enough naive individuals for at least one of them to acquire infection (for infectious diseases), or to mate successfully (for any species undergoing sexual reproduction).

(ii) **Compatibility**: Both invasive species and EIDs must be capable of surviving in their new environment to the point of reproduction, and then of reproduction that supports stable or expanding populations. The capacity of an invading species to reproduce in the invaded environment is often measured as the intrinsic growth rate of the population ($r$, which is a time-based metric) in invasion science and the basic reproduction number ($R_0$, which is a generation-based metric) in epidemiology. For persistence (i.e. naturalization) of invasive populations or EIDs, they must be compatible with 'environmental' conditions (including quantities such as host

population size and density) to the extent that $r$ is positive and $R_0$ is greater than unity [15]. Whether or not an introduced organism becomes naturalized or invasive depends, to a great extent, on the eco-evolutionary experience of the introduced species and the recipient community. Eco-evolutionary experience describes the historical exposure of an organism to biotic interactions over evolutionary timescales [48,49], and emphasizes the role of traits selected for in previous environments (pre-adaptations), within both introduced and resident species, in driving the establishment success and adaptability of introduced species. In other words, eco-evolutionary experience determines the ease with which an invader can integrate into novel ecological contexts, and pre-adaptations are crucial determinants of a species' invasiveness and a community's invasibility [48–51]. Continuing evolutionary change of invading species is commonplace [52], and often involves admixture (intraspecific hybridization between previously allopatric populations) or hybridization between closely related species (e.g. [53]). Such genetic recombination often leads to enhanced performance by invasive populations due to heterosis and hybrid vigour [54]. However, many invasive species adapt in the absence of admixture or hybridization [54,55], resulting in traits that increase their performance. For example, invasive species may undergo rapid evolution in traits related to dispersal (e.g. [56]) and much insight has been gathered on such adaptations by identifying candidate genes underlying them. Adaptive emergence of EIDs for transmissibility of animal pathogens to or among humans explicitly requires genetic change, by mutations and recombination events [10,11]. However, as for non-disease-causing invasive species, pathogens and disease vectors continue to evolve and adapt to new environments into which they have been introduced, enhancing $R_0$ within the invaded environments [57,58]. For pathogens of animals and humans, evolution towards increased $R_0$ typically involves trade-offs between traits of transmission (higher pathogen loads mean more efficient transmission when contact is made between infected and naive hosts) and virulence (higher pathogen loads mean greater morbidity/mortality and reduced contact rates between infected and naive hosts) [59]. Such evolutionary processes are, however, highly idiosyncratic among pathogens that are transmitted by different routes [60] and among different populations [61]. Genetic changes may also permit invasive species and EIDs to persist long-term and not undergo 'boom and bust' which may occur for a range of reasons, including depletion of resources [62,63].

(iii) **Environment**: Environmental conditions determine whether a recipient location provides a suitable niche for species to establish and spread. Abiotic factors including climate (e.g. temperature, rainfall/humidity), and substrate qualities are key to whether introduced species can survive. Biotic factors, ranging from host population size, density and connectedness, and nutritional resources through 'enemies' (predators, parasites, pathogens, competitors and, for microorganisms, immunity and cross-immunity) to more complex community interactions, will determine whether introduced species can survive and reproduce [32,64]. When biogeographic barriers are breached by human action, species may be introduced to ecological niches that are suitable for their survival and reproduction and which also provide an 'enemy-free' space that further permits their establishment and spread. For this reason, the realized niche of species may be much larger in their introduced ranges than their native ranges [65]. The same is true of EIDs when they are introduced into an immunologically naive population [15]. While evolutionary change in invading species may alter the compatibility of the invading species with the invaded environment, environmental change may facilitate invasions by creating new suitable niches for invading species without the need for evolutionary change. Human disturbance of natural communities, ranging from replacement of natural vegetation with agricultural systems to more subtle changes, can make them more vulnerable to invasive species [66,67]. Such changes have similar effects on the process of emergence of infectious diseases in both wildlife and livestock [68]. Current and future global change (climate, biodiversity, landscape/land-use change, including urbanization) are likely to facilitate both disease emergence and biological invasions, while some sudden and unpredictable environmental fluctuations may inhibit invasions [69,70].

In the above section, we have separated geographical, compatibility and environmental barriers, but they are often interdependent in influencing invasion/emergence ($r$ and $R_0$ depend on both compatibility and environment). Even when not mutually dependent, they act together. For example, environmental change (such as altered land use) can bridge the 'geographical' contact barrier between animal pathogens and humans, as is the case for Nipah virus [71]. Environmental changes also drive evolutionary changes that may alter the eco-evolutionary experience of potential invaders and

potentially invaded communities. Issues of global spread of species and global environmental changes that drive disease emergence directly and indirectly (via non-disease-causing invasive species) underline the need for a One Health approach [23].

## 2.3. Similar methods

Risk analysis is a key management approach for both applied epidemiologists and invasion biologists. In this section, we focus primarily on risk assessment and return to discuss risk management later. Risk assessment is applied to help develop policies in anticipation of, and in response to, disease emergence events and biological invasions. To support these risk assessments, both disciplines aim to identify qualities (traits or syndromes) that (i) make species 'invaders' or 'emergers' (e.g. [72–74]), (ii) make source environments more likely to yield them (e.g. [74]), and (iii) render receiving environments susceptible or resistant to invaders or emerging pathogens [75]. Modelling is used in both invasion science and epidemiology to elucidate biological processes, predict establishment and spread, to support risk assessment and to assess effectiveness of interventions. The same 'top-down' (correlative, e.g. statistical models, ecological niche models and machine learning) and 'bottom-up' (mechanistic, e.g. dynamic simulation models, network analysis, individual-based models) methods are used for predicting the possible current and future extent of EIDs and invasive species [32,76]. Disease modelling methods used by epidemiologists would, of course, be directly relevant to modelling all types of infectious diseases, including those that affect species other than vertebrates, including plant pathogens [77]. Methods for monitoring invasive species, including active field surveillance and citizen science-based passive surveillance, have much in common with methods used to monitor risks from emerging zoonoses and vector-borne diseases in the environment [78–81]. Similar sampling designs are used and their implementation in target regions or sentinel sites is often determined by similar criteria, such as likely spread patterns predicted by species distribution and spread models, and occurrence of locations where impact may be greatest (e.g. [82]). In both disciplines, molecular approaches are used to confirm species identities and for source attribution [83,84], and both are exploring Earth observation data as proxies for potential occurrence of invaders [85], or risk from EIDs [86].

# 3. Useful differences: opportunities for synergies

## 3.1. Differences in scope

From an invasion biology perspective, EIDs are idiosyncratic in two ways. First, many important EIDs affecting humans and domesticated animals are obligate parasites of vertebrates [5], which means that consideration of the host population is paramount to predictive modelling and assessing impacts and risk. Parasitic species and microorganisms thus comprise a special subset of invasive species. For EIDs and parasitic invasive species, spread into naive populations may be rapid from the point of introduction to an epidemic, provided there is sufficient availability of naive hosts. To a first approximation, spread will not occur if the frequency of contact with naive hosts is below a threshold level. For microorganisms transmitted directly among humans, the patterns and extent of spread (equivalent to the 'invasive range') are mostly determined by characteristics of the human population and microorganism and not directly by the environment. The persistence of transmission cycles of microorganisms following spread (i.e. endemicity) depends on the details of the transmission characteristics of the microorganism and of the host population. As for non-infectious invasive species, emerging infections may boom and bust but usually due to mechanisms associated with the availability of susceptible hosts, through either reduction in the host population by a highly pathogenic EID or the development of immunity to the emerging pathogen in the host population [87].

Second, the causal organisms of EIDs (viruses, bacteria, fungi, protozoa and helminths) and vectors (particularly insects) are, for the most part, at the 'small and fast' end of the spectrum of invasive species, i.e. they have very small size and their generation time is often (but not always) short (days to months). By contrast, generation times may be years to decades for organisms like invasive trees. Notably, few invasive plants have reached their broad-scale climatic limits in their new ranges even centuries after introduction (e.g. [88–90]). Given the ease of accidental long-distance movement by human agency, microorganisms are likely to be common as invasive species of natural systems globally, although data on the occurrence of such events are very limited. Furthermore, due to their extremely

short generation times, compared to many invasive plant species for example, they have greater capacity to adapt genetically to new environments. Despite this, and compared to their focus in EID epidemiology, microorganisms remain understudied in invasion biology due to a range of factors including difficulties with isolation or culturing, poorly known biogeography and therefore their native versus non-native status, and difficulties in detecting and ascribing impacts to the causative agent (e.g. [91]).

The first difference described above could be thought of as a limit on the scope of direct synergies in models used and the number of 'invasive EIDs' that may lend themselves to direct collaborations between invasion biologists and epidemiologists. However, clearly some invasive species are parasites or pathogens, and for these, the expertise of EID epidemiologists would enrich invasion biology. Furthermore, this apparent idiosyncrasy does not mean that invasion biologists cannot profit from modelling approaches developed in EID epidemiology. The second difference is of interest because the larger size (which makes their detection and enumeration easier) and longer generation times of many invasive species have meant that the demographic processes and community ecology of invasions have been more readily studied. Epidemiologists tend to use relatively simple criteria-led approaches or species distribution models to assess whether, and to what extent, invasion by pathogens and vectors may occur now and in the future (e.g. [92]). The approach to understanding the processes of introduction–naturalization–invasion used by invasion biologists has made it easier to describe and understand individual invasion processes [32]. This approach could be used to enhance risk assessment for EIDs, particularly those that are vector-borne and those that are zoonoses associated with wildlife, as all of the factors involved in these processes may determine the speed, trajectory and impact of EIDs as well as invasive species.

Factors that make species more successful invaders have been studied in invasion science since the 1980s (including using approaches of comparing native with invasive species, and invasive alien with alien-but-not-invasive species [93]), but only more recently by epidemiologists interested in emerging diseases [72,94,95]. Consequently, the elucidation of traits of invasiveness and invasibility and the recognition that these traits of invaders and invaded communities interact to permit or prevent invasions [96] is generally much richer than for EIDs. Studies in invasion science have led to concepts of traits that permit invaders to be more successful in certain environments (e.g. 'urban winner' species [97]), and ordination-type methods for classifying communities in terms of their invasibility (e.g. 'periodic tables of niches' [98]). All of these could be a focus for direct knowledge transfer from invasion science to those assessing risk of zoonotic EIDs and arthropod vectors, and for conceptual exploration of their application to assessing risk of all EIDs. Ultimately, this may significantly enhance our understanding of the different components of the emergence/invasion systems allowing more effective prevention and control strategies.

## 3.2. Differences in risk management methods

As invasion biologists and epidemiologists have practical objectives of reducing impacts of the species that are their focus (by prevention, eradication, containment, control or impact reduction), sharing of tools, methods and activities that facilitate these objectives may have considerable value. This subject is worthy of a review in its own right—the following are simply examples.

While risk assessment of an anticipatory nature is very similar in the fields of infectious disease epidemiology and biological invasions, there are differences when risk management is conducted in the face of invasions or EIDs. In invasion science, risk management addresses the consequences of inaction by estimating the 'invasion debt', primarily of existing introduced species [99]. This approach could be readily adapted to risk management practices for EIDs. Those responsible for managing invasions use a range of tools, such as eradographs, to visualize the impacts of interventions to control geographical spread [100] and identification of management-specific switch points in control programmes that determine if and when management objectives should be changed [101].

Field surveillance/monitoring is conducted for both EIDs (particularly when these are zoonoses or vector-borne) and invasive species [102,103], and it may be practical and economical to develop combined field surveillance programmes. For example in Canada, south-to-north invasion of tick and fly vectors and of vector-borne pathogens of human and livestock health significance is occurring or a threat [102,103]. While the vectors and vector-borne diseases of livestock may not have human health importance, surveillance may use methods and/or locations similar enough for collaborations in field surveillance to be logical. Molecular methods are mainstream in identifying microbial pathogens in infectious disease surveillance programmes, but these methods are almost entirely used for identifying pathogens and comparisons to identify disease clusters or to attribute sources [104]. More detailed

molecular analytical approaches are used in invasion science to understand invasion dynamics, such as underlying propagule pressure [105], landscape-scale dispersal patterns and rates [106], or to reconstruct invasion history and pathways [83,107]. These approaches may assist risk assessment and policies for management [108], while analysis of environmental DNA using meta DNA barcoding can assist in detecting any species (non-infectious invasive and EIDs) during transport, thereby aiding in preventing introductions from occurring [109]. While molecular approaches are often used to identify the provenance of source populations of invading populations and EIDs [84], they can also provide information relevant to biological control of invasive populations, for example, identifying the native regions where the prospects of identifying co-evolved biological control agents are more likely [110]. All of these more detailed approaches could be more widely implemented in the field of EID surveillance.

Passive citizen science methods of collecting information on species distributions are used both in public health and in ecology. In ecology, the object is monitoring of biogeography and global biodiversity information (e.g. eButterfly—http://www.e-butterfly.org/ and iSpot—https://www.ispotnature.org/). However, in public health, these methods have been developed to the point where data are systematically collected and analysed in national surveillance programmes to provide early warning of emerging vector-borne diseases allowing rapid responses [111]. Because most invasion science does not (directly) address human health issues, funding is probably much more difficult to mobilize for work on invasions than for EIDs. This means that cheaper means must be sought to detect new introduced species than can be implemented for EIDs. Nonetheless, the experience of public health epidemiologists in this area may benefit the field of invasion science, and epidemiologists may benefit from incorporating more cost-efficient methods developed in invasion biology.

In public health epidemiology, the need for rapid, specific and sensitive methods to detect clusters of disease cases as the first sign of an outbreak has led to a revolution in molecular and bioinformatics methods (particularly whole-genome sequencing and analysis) for species identification [104]. Given the potential for EID epidemics to arise rapidly, there has been considerable effort in public health to implement these molecular methods into programmes that systematically identify and control EIDs [111]. These complement data-driven international efforts to detect EID events including the joint WHO-OIE-FAO Global Early Warning System (GLEWS) for health threats and emerging risks at the human–animal–ecosystems interface (http://www.glews.net/), active detection of possible EID events via international media reports by the Global Public Health Intelligence Network (GPHIN), and passive detection of EID events by interested, voluntarily participating public health, infectious disease, veterinary, microbiology and academic experts in systems such as Promed (https://www.promedmail.org/) and Health Map (https://www.healthmap.org/en/) [111].

In general, control methods for EIDs (e.g. vaccines and quarantine) and invasive species (e.g. plant removal) are highly idiosyncratic, even if at first sight (such as chemical control of insects), they may seem very similar. However, despite these clear differences, prevention and control programmes for both EIDs and invasive species share the potential for interactions with the public to be crucial for programmes to succeed. Public trust and engagement (for example, in terms of personal and environmental impact, privacy/data-security, land ownership and access) may be essential for successful prevention and control [112]. Collaborations in developing procedures for public engagement may be very fruitful.

The often-rapid nature of disease emergence requires quick mobilization of expertise, and resources, including funding and personnel. The immediate relevance of EIDs to humans has united global efforts to counter them, and this has resulted in national and international networks of public health organizations coordinated (in the case of international outbreaks) by WHO. By contrast, calls for unity over invasions (e.g. [113]) have so far largely failed to produce effective agencies. There is, for example, no equivalent to public health organizations such as US Centers for Disease Control and Prevention, the Public Health Agency of Canada and the European Centre for Disease Prevention and Control that has prime responsibility for the detection and control of invasive species. This contrast is probably due to a number of reasons, including the local, regional or national (versus international) scope of many invasions, the often long time lag between biological invasions and detected impacts, and the generally slower nature of invasions, which together result in fractionated efforts that may be ineffective. Responsibilities for coordinating responses to EIDs of public health significance always lie with public health organizations, but responsibilities for responding to invasive species vary depending on the impact or location of the invasive species and may be organizations responsible for agriculture, fisheries, environment, natural resources, transport or local government entities [114].

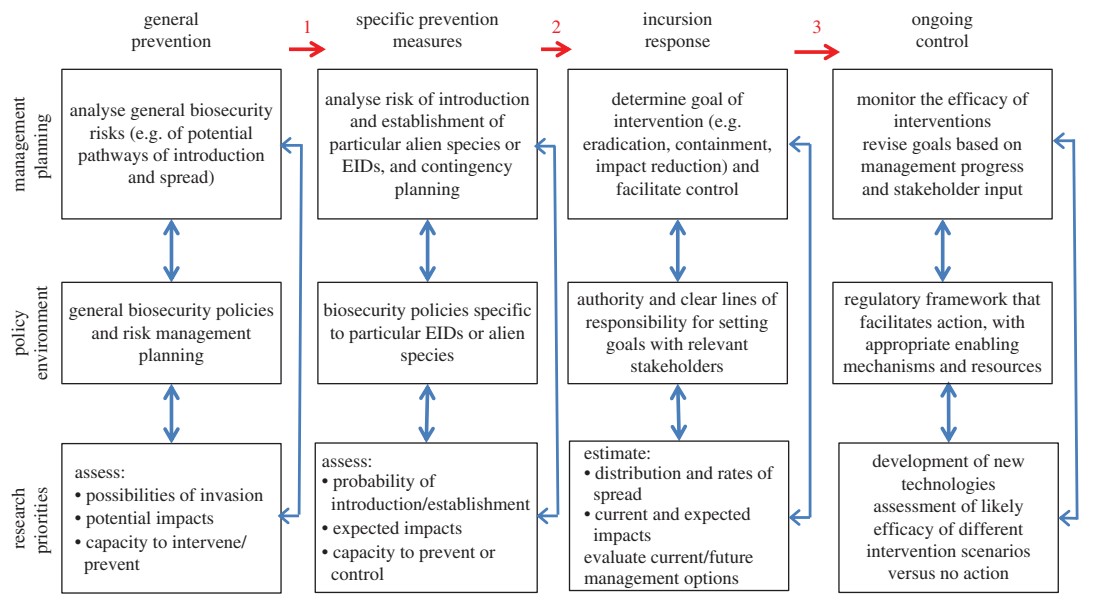

**Figure 4.** A One Health approach to the management of EIDs and biological invasions. The continuum of possible invasion/EID management functions, their policy or programme objectives, and the research activities that support their development are shown. The rows of boxes represent the different fields involved in responding to EIDs and biological invasions: management programmes, policy development and scientific research. The columns represent different stages of response to EIDs and invasions and how, as indicated by numbered red arrows, emphasis may change from general research into risks of EIDs and invasions, to focus on: (1) particular potential threats; (2) species/EIDs detected as invading; and (3) ongoing management of EIDs and invasive species.

# 4. Where to from here?

Invasion science and epidemiology in the context of EIDs represent both applied and basic sciences. Both disciplines are involved in, and are informed by, fundamental research, and both have clear objectives and mandates to minimize negative impacts on society and the environment. The science is applied to the development of management plans along a continuum of points of potential anticipatory and responsive actions (figure 4). These functions are, to a greater or lesser extent, already undertaken independently by those involved in the study, prevention and control of EIDs and biological invasions. However, we advocate for a strong, collaborative One Health approach in these actions that integrates across human, animal and environmental health, including both invasion biology and epidemiology in the field of EIDs.

There are clear opportunities for immediate collaboration:

(1) Predictive modelling: Modelling of dispersal, introduction and spread of EIDs and invasive species would be a relatively simple point of collaboration as the objectives are similar. This would be applicable to all parts of the continuum of management functions, but particularly to anticipating the risk of invasions and EIDs.

(2) Monitoring of EIDs and invasions: International scanning, as conducted for EIDs, could be readily applied to biological invasions as has been proposed previously [115]. The increasing availability of 'big data' to support detection and monitoring of EIDs and biological invasions, and the challenges of analysing these data to provide intelligence, is a particularly needed avenue of collaborative action and research [115,116]. Collaborative monitoring (and more systematic surveillance) for EIDs and invasive species at points of entry, monitoring in field studies (including assessing indirect effects of invasions on health), and collaboration on the development and application of molecular methods for detection and demographic analysis of populations of invasive species and EIDs are all areas where synergistic activities could increase efficiency.

(3) Management of invasions and EIDs: Given the transferable skill sets between those involved in EID and invasive species management, and the possibilities for synergies between the fields, collaboration across the range of management activities could be very advantageous, recognizing that the responsibilities for managing biological invasions and EIDs may be distinct.

We are not aware of examples where the application of epidemiology to biological invasions or invasion biology to EIDs has resulted in improved outcomes in terms of prevention or control. However, the application of epidemiological modelling may well have contributed to understanding patterns of spread of chytridiomycosis in amphibians, which is very likely a transmissible disease (e.g. [117]). Similarly, while the emerald ash borer beetle and WNV (both invasive species) arrived almost simultaneously at the US–Canada border, to date practical spread modelling of the emerald ash borer as conducted by invasion biologists [118] has only been matched by theoretical mathematical modelling of the spread of WNV [119]. Furthermore, there is no cross-talk between those responsible for predicting spread and responding to the emerald ash borer (https://www.nrcan. gc.ca/forests/fire-insects-disturbances/top-insects/13377), and those responsible for responding to WNV (https://www.canada.ca/en/public-health/services/diseases/west-nile-virus/surveillance-west-nile-virus.html), when it is recognized that changes in biodiversity (such as those occurring as a consequence of the invasion of emerald ash borer) may have impacts on risk from WNV [120]. The lack of integration of these responses provides a clear example of a missed opportunity to benefit from the One Health framework.

Throughout, collaborations need to be win–win for epidemiologists and invasion biologists, and they need to be enabled. 'Soft' collaborations within the academic context would be the easiest to set up, and may only require simple encouragement (e.g. joint seminars, learning exchanges or workshops). More solid collaborations on joint projects, such as proposed for 'Global networks for invasion science' [121], would require collaborative funding opportunities. Possibly the most enabling step would be the development of common collaborative programmes founded on common policy initiatives of national and international organizations responsible for managing EIDs and biological invasions. In the One Health field, this has begun with the animal health, human health and food security organizations working collaboratively on the FAO/OIE/WHO Tripartite Collaboration on antimicrobial resistance (AMR: http://www.who.int/foodsafety/areas_work/antimicrobial-resistance/ tripartite/en/). While human health has a UN organization, the WHO, that provides international leadership and coordination on EIDs, there is currently no analogous body for invasion science. The United Nations Environment Programme (UNEP) and IUCN's Invasive Species Specialist Group (ISSG) may be two of the most promising institutional bodies that could facilitate interactions between invasion biologists and epidemiologists (and their organizations).

## 5. Conclusion

The fields of invasion science and EID epidemiology share the challenge of the increasing numbers of invasions and EIDs with no evidence of saturation. Invasions and EIDs involve similar biological processes, may be intrinsically linked biologically and by human activity, are addressed by scientists with similar skills and objectives, and are being driven by the same global changes. Invasions by non-pathogenic organisms can also have important impacts on human health. They are therefore both part of the One Health concept and require a One Health approach to minimize their negative impacts on humanity. We have identified exciting opportunities for synergies between the fields of invasion science and EID epidemiology and call for greater collaboration to benefit humanity.

Data accessibility. There are no data associated with this review article.

Authors' contributions. N.H.O. led writing and all authors contributed to both the concept and the text. All authors gave final approval for publication.

Competing interests. The authors declare no competing interests.

Funding. N.H.O. is funded by the Public Health Agency of Canada. J.R.C.P. is funded by SACEMA, which is a DST-NRF Centre of Excellence. S.J.D., C.H., S.K., J.J.L.R., J.M., D.M.R., W.-C.S. and J.R.U.W. acknowledge funding from the DST-NRF Centre of Excellence for Invasion Biology (CIB). Development of this article was supported by a joint SACEMA-CIB workshop funded by SACEMA and hosted on 9–12 October 2017 at Stellenbosch University.

Acknowledgements. We thank the following participants in the joint SACEMA-CIB workshop who provided diverse insights into the issues of collaborations between invasion biologists and EID epidemiologists: Dr Gbenga Abiodun, Department of Mathematics, University of the Western Cape; Dr John Grewar, Western Cape Department of Veterinary Services; Dr John Hargrove, SACEMA, Stellenbosch University; Prof. Charles Kadzere, Eastern Cape Department of Rural Development and Agrarian Reform; Dr Hassan Mahomed, Cape Metropolitan Health District, Western Cape Government; Dr Ziyanda Majokweni, Poultry Disease Management Agency, South African Poultry Association; Ms Zinhle Mthombothi, SACEMA, Stellenbosch University.

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
