## [Reviewer comments · Royal Society Open Science]

Review History

RSOS-181577.R0 (Original submission)

Review form: Reviewer 1

Is the manuscript scientifically sound in its present form?

Yes

Are the interpretations and conclusions justified by the results?

Yes

Is the language acceptable?

Yes

Is it clear how to access all supporting data?

Not Applicable

Do you have any ethical concerns with this paper?

No

Have you any concerns about statistical analyses in this paper?

No

Recommendation?

Accept with minor revision (please list in comments)

Comments to the Author(s)

A position paper/review that compares and contrasts the fields and efforts in the control of invasive biologicals (plants, insects, others), and those in the control of emerging infectious diseases. The connections between those fields are still largely distinct, and populated by different researchers, but they make a good case that there are many connections in the introduction, initial establishment, emergence and spread of the two types of invasives. Many of the same skills are used in the two fields, and in modeling the different steps involved, indicating that the disciplines can share skills and models. They build this around the so-called One Health model, which is reasonable, and they include some well thought out/diagrammatic models of the steps involved in analysis and control of each type of invasion/emergence. Overall a useful contribution to the fields involved, and it will be interesting to see if the cooperation between the fields can be accomplished.

A few issues to consider;

- 1) They seem to leave out or at least not explicitly include an important component of the One Health effort - the medical communities (human and veterinary). Perhaps the invasive studies do not obviously connect up, but the emerging diseases would do?
- 2) A few jargon words that I am not immediately sure of the definition - disservices? Maybe those can be defined or a glossary included.
- 3) They do not explicitly seem to show how the fields are connected by invasive vectors (mosquitoes, ticks, others) and that provide a pathway for the diseases that take advantage of those to spread into new areas.
- 4) The term pre-adaptation is unclear to me, and perhaps not really an accurate definition of certain mutations. Not sure what is meant by "eco-evolutionary experience". Mutations arise and selected in an environment, which may favor spread in another environment - not sure if that is pre-adaptation as it implies some foreshadowing of future selections in other environments?
- 5) Page 6 - ii) - Compatibility? For disease emergence this would also include population size, density and connectedness?
- 6) Page 6 - Would be "as for non-disease-causing invasive..."
- 7) Page 8 - Not sure that EIDs are typically obligate parasites, or of vertebrates. Many could be opportunistic; also could equally well emerge in invertebrates (or plants or prokaryotes)?
- 8) Page 9 - "relatively recently" is cited to a 2001 paper - almost a generation ago.
- 9) Page 9 - determining parameters of the different components of the emergence/invasion systems would be needed for proper control strategies?
- 10) Page 10 - what is the data showing that funding is "orders of magnitude" less for invasion studies.
- 11) Page 10 - why is only the European CDCD cited? Most counties have organizations with similar responsibilities that might be mobilized? What are the organizations that address invasive transmission?
- 12) Figure 3 - This is an odd diagram and unclear - perhaps the barriers need to be extended across the different columns to make it clear what it is showing.

Review form: Reviewer 2

Is the manuscript scientifically sound in its present form?

Yes

Are the interpretations and conclusions justified by the results?

Yes

Is the language acceptable?

Yes

Is it clear how to access all supporting data?

Not Applicable

Do you have any ethical concerns with this paper?

No

Have you any concerns about statistical analyses in this paper?

No

Recommendation?

Major revision is needed (please make suggestions in comments)

Comments to the Author(s)

The paper you have submitted says, in the title, that it's a call for closer collaboration and, on p. 3 line 52, that it's a review. In p. 3 lines 46-7 you say the ideas are not new. In my view, your paper is a very useful and thorough survey of some relevant science. The science is embedded in phrases that say that both invasionists and epidemiologists 'require' (p 12 line 7) a joint One Health approach. You need to say, both in the title and early on, that the paper is both a call and a review and to connect the two more clearly. In particular you need clearer arguments that invasion scientists and epidemiologists would produce better results than working separately and, also, why those two disciplines. Are not e.g. socio-economic factors important in the management of EIDs and invasions? Is there not an organisation and journal called EcoHealth working on that? See vol 15 part 2, June 2018 for a symposium on EIDs and economics. Maybe you should also look at EPPO (The European and Mediterranean Plant Protection Organization) and give more attention to plant diseases.

I think though that the chief weakness is that you don't say enough about One Health as an organisation. What sort of a body is it? What does it do? Has it had any results that have had an important effect on the management of human and /or animal diseases? Its web site gives the impression of a sleepy organisation just running meetings and not keeping its web site up to date. The arguments connecting One Health and a possible new UN (or other) organisation with your survey could be set out less vaguely, with better examples of how a One Health approach could help. Maybe you could be explicit about some situations where a One Health approach would clearly help and others where it would not be useful. For instance, could epidemiology help in managing pine invasions in the fynbos and, if so, how? And conversely, invasion science and managing AIDS in Cape Town?

The bulk of your paper is the survey of the similarities and differences between epidemiology and invasion science. Most of it is excellent and would be very useful to both communities. You put your over-view into a series of figures of a type that I personally don't find helpful but many people do. One Health is based on combing the three fields called Human, Animal and Environmental in Fig 1, but the joint set of those three is empty in fig 1 (apart from an arrow passing through). In Fig 2 the boundary of Animal is oddly placed. Fig 3 shows a scheme

following one devised by Dr Richardson with barriers and there is an assertion that the first barrier to cross is a geographical one, as it was with WNV. But I would have thought that most EIDs have first to cross a species barrier, as HIV had to jump from chimpanzee to man or ebola from fruit bats (maybe) with no geographical barrier involved. Fig 4 gets as far as management planning with no management action,

Which brings me to the most disappointing section, on p. 9 'B) Differences in risk management methods' which is a classified reading list not a review and, in particular no results, no examples of real management. As the obvious difference between EIDs and classical biological invasions is the way they have to be managed, this is a major gap that should be filled. This section needs to be greatly expanded.

The other areas that I think are neglected are, as mentioned above, plant pathogens and, quite different, insect parasitoids whether introduced for biological control or, occasionally, introduced pests. And maybe other insect parasites such as leaf-miners.

I would hope that such revisions would make a much more compelling case.

Decision letter (RSOS-181577.R0)

04-Jan-2019

Dear Dr Ogden,

The editors assigned to your paper ("Emerging infectious diseases and biological invasions – a call for closer collaboration in science and management") have now received comments from reviewers. We would like you to revise your paper in accordance with the referee and Associate Editor suggestions which can be found below (not including confidential reports to the Editor). Please note this decision does not guarantee eventual acceptance.

Please submit a copy of your revised paper before 27-Jan-2019. Please note that the revision deadline will expire at 00.00am on this date. If we do not hear from you within this time then it will be assumed that the paper has been withdrawn. In exceptional circumstances, extensions may be possible if agreed with the Editorial Office in advance. We do not allow multiple rounds of revision so we urge you to make every effort to fully address all of the comments at this stage. If deemed necessary by the Editors, your manuscript will be sent back to one or more of the original reviewers for assessment. If the original reviewers are not available, we may invite new reviewers.

- Data accessibility

<http://datadryad.org/submit?journalID=RSOS&manu=RSOS-181577>

- Competing interests

- Authors' contributions

- Acknowledgements

- Funding statement

on behalf of Prof Kevin Padian (Subject Editor)
 openscience@royalsociety.org

Comments to Author:

Reviewers' Comments to Author:
 Reviewer: 1

Comments to the Author(s)

A position paper/review that compares and contrasts the fields and efforts in the control of invasive biologicals (plants, insects, others), and those in the control of emerging infectious diseases. The connections between those fields are still largely distinct, and populated by different researchers, but they make a good case that there are many connections in the introduction, initial establishment, emergence and spread of the two types of invasives. Many of the same skills are used in the two fields, and in modeling the different steps involved, indicating that the disciplines can share skills and models. They build this around the so-called One Health model, which is reasonable, and they include some well thought out/diagrammatic models of the steps involved in analysis and control of each type of invasion/emergence. Overall a useful contribution to the fields involved, and it will be interesting to see if the cooperation between the fields can be accomplished.

A few issues to consider;

- 1) They seem to leave out or at least not explicitly include an important component of the One Health effort - the medical communities (human and veterinary). Perhaps the invasive studies do not obviously connect up, but the emerging diseases would do?
- 2) A few jargon words that I am not immediately sure of the definition - disservices? Maybe those can be defined or a glossary included.
- 3) They do not explicitly seem to show how the fields are connected by invasive vectors (mosquitoes, ticks, others) and that provide a pathway for the diseases that take advantage of those to spread into new areas.
- 4) The term pre-adaptation is unclear to me, and perhaps not really an accurate definition of certain mutations. Not sure what is meant by "eco-evolutionary experience". Mutations arise and selected in an environment, which may favor spread in another environment - not sure if that is pre-adaptation as it implies some forshadowing of future selections in other environments?
- 5) Page 6 - ii) - Compatibility? For disease emergence this would also include population size, density and connectedness?
- 6) Page 6 - Would be "as for non-disease-causing invasive..."
- 7) Page 8 - Not sure that EIDs are typically obligate parasites, or of vertebrates. Many could be opportunistic; also could equally well emerge in invertebrates (or plants or prokaryotes)?
- 8) Page 9 - "relatively recently" is cited to a 2001 paper - almost a generation ago.

- 9) Page 9 - determining parameters of the different components of the emergence/invasion systems would be needed for proper control strategies?
- 10) Page 10 - what is the data showing that funding is "orders of magnitude" less for invasion studies.
- 11) Page 10 - why is only the European CDCD cited? Most counties have organizations with similar responsibilities that might be mobilized? What are the organizations that address invasive transmission?
- 12) Figure 3 - This is an odd diagram and unclear - perhaps the barriers need to be extended across the different columns to make it clear what it is showing.

Reviewer: 2

Comments to the Author(s)

The paper you have submitted says, in the title, that it's a call for closer collaboration and, on p. 3 line 52, that it's a review. In p. 3 lines 46-7 you say the ideas are not new. In my view, your paper is a very useful and thorough survey of some relevant science. The science is embedded in phrases that say that both invasionists and epidemiologists 'require' (p 12 line 7) a joint One Health approach. You need to say, both in the title and early on, that the paper is both a call and a review and to connect the two more clearly. In particular you need clearer arguments that invasion scientists and epidemiologists would produce better results than working separately and, also, why those two disciplines. Are not e.g. socio-economic factors important in the management of EIDs and invasions? Is there not an organisation and journal called EcoHealth working on that? See vol 15 part 2, June 2018 for a symposium on EIDs and economics. Maybe you should also look at EPPO (The European and Mediterranean Plant Protection Organization) and give more attention to plant diseases.

I think though that the chief weakness is that you don't say enough about One Health as an organisation. What sort of a body is it? What does it do? Has it had any results that have had an important effect on the management of human and /or animal diseases? Its web site gives the impression of a sleepy organisation just running meetings and not keeping its web site up to date. The arguments connecting One Health and a possible new UN (or other) organisation with your survey could be set out less vaguely, with better examples of how a One Health approach could help. Maybe you could be explicit about some situations where a One Health approach would clearly help and others where it would not be useful. For instance, could epidemiology help in managing pine invasions in the fynbos and, if so, how? And conversely, invasion science and managing AIDS in Cape Town?

The bulk of your paper is the survey of the similarities and differences between epidemiology and invasion science. Most of it is excellent and would be very useful to both communities. You put your over-view into a series of figures of a type that I personally don't find helpful but many people do. One Health is based on combing the three fields called Human, Animal and Environmental in Fig 1, but the joint set of those three is empty in fig 1 (apart from an arrow passing through). In Fig 2 the boundary of Animal is oddly placed. Fig 3 shows a scheme following one devised by Dr Richardson with barriers and there is an assertion that the first barrier to cross is a geographical one, as it was with WNV. But I would have thought that most EIDs have first to cross a species barrier, as HIV had to jump from chimpanzee to man or ebola from fruit bats (maybe) with no geographical barrier involved. Fig 4 gets as far as management planning with no management action,

Which brings me to the most disappointing section, on p. 9 'B) Differences in risk management methods' which is a classified reading list not a review and, in particular no results, no examples of real management. As the obvious difference between EIDs and classical biological invasions is the way they have to be managed, this is a major gap that should be filled. This section needs to be greatly expanded.

The other areas that I think are neglected are, as mentioned above, plant pathogens and, quite different, insect parasitoids whether introduced for biological control or, occasionally, introduced pests. And maybe other insect parasites such as leaf-miners.

I would hope that such revisions would make a much more compelling case.

Author's Response to Decision Letter for (RSOS-181577.R0)

See Appendix A.

RSOS-181577.R1 (Revision)

Review form: Reviewer 2

Is the manuscript scientifically sound in its present form?

Yes

Are the interpretations and conclusions justified by the results?

Yes

Is the language acceptable?

Yes

Is it clear how to access all supporting data?

Not Applicable

Do you have any ethical concerns with this paper?

No

Have you any concerns about statistical analyses in this paper?

No

Recommendation?

Accept as is

Comments to the Author(s)

Thank you for your careful responses. I have just a couple of suggestions for wording. In lines 37 & 38 you say necessary and requires which have a touch of hype about them which may make people sceptical. In lines 103, 881 you say actor which, in English English, is almost always someone on a stage or in a film. Your usage may be commoner in other English speaking countries but for a London based publication I would suggest participant or agent (or even colleague or collaborator).

Decision letter (RSOS-181577.R1)

14-Feb-2019

Dear Dr Ogden:

On behalf of the Editors, I am pleased to inform you that your Manuscript RSOS-181577.R1 entitled "Emerging infectious diseases and biological invasions – a call for a One Health collaboration in science and management" has been accepted for publication in Royal Society Open Science subject to minor revision in accordance with the referee suggestions. Please find the referees' comments at the end of this email.

The reviewers and Subject Editor have recommended publication, but also suggest some minor revisions to your manuscript. Therefore, I invite you to respond to the comments and revise your manuscript.

- Ethics statement

- Data accessibility

If you wish to submit your supporting data or code to Dryad (<http://datadryad.org/>), or modify your current submission to dryad, please use the following link:
<http://datadryad.org/submit?journalID=RSOS&manu=RSOS-181577.R1>

- Competing interests

- Authors' contributions

- Acknowledgements

- Funding statement

Because the schedule for publication is very tight, it is a condition of publication that you submit the revised version of your manuscript before 23-Feb-2019. Please note that the revision deadline will expire at 00.00am on this date. If you do not think you will be able to meet this date please let me know immediately.

Supplementary files will be published alongside the paper on the journal website and posted on

the online figshare repository (<https://figshare.com>). The heading and legend provided for each supplementary file during the submission process will be used to create the figshare page, so please ensure these are accurate and informative so that your files can be found in searches. Files on figshare will be made available approximately one week before the accompanying article so that the supplementary material can be attributed a unique DOI.

on behalf of Prof Kevin Padian (Subject Editor)
openscience@royalsociety.org

Reviewer comments to Author:
Reviewer: 2

Comments to the Author(s)

Thank you for your careful responses. I have just a couple of suggestions for wording. In lines 37 & 38 you say necessary and requires which have a touch of hype about them which may make people sceptical. In lines 103, 881 you say actor which, in English English, is almost always someone on a stage or in a film. Your usage may be commoner in other English speaking countries but for a London based publication I would suggest participant or agent (or even colleague or collaborator).

Author's Response to Decision Letter for (RSOS-181577.R1)

See Appendix B.

Decision letter (RSOS-181577.R2)

18-Feb-2019

Dear Dr Ogden,

I am pleased to inform you that your manuscript entitled "Emerging infectious diseases and biological invasions – a call for a One Health collaboration in science and management" is now accepted for publication in Royal Society Open Science.

You can expect to receive a proof of your article in the near future. Please contact the editorial

office (openscience_proofs@royalsociety.org and openscience@royalsociety.org) to let us know if you are likely to be away from e-mail contact. Due to rapid publication and an extremely tight schedule, if comments are not received, your paper may experience a delay in publication.

on behalf of Prof Kevin Padian (Subject Editor)
openscience@royalsociety.org

Appendix A

Dear Mr Dunn and Prof. Padian, many thanks to you and your reviewer for the comments on our article. We have responded to these as detailed below and hope the article is now acceptable to you. Page and line numbers refer to the article with track changes visible.

Many thanks for your help with this.

Nick Ogden

Reviewer: 1

Comments to the Author(s)

A position paper/review that compares and contrasts the fields and efforts in the control of invasive biologicals (plants, insects, others), and those in the control of emerging infectious diseases. The connections between those fields are still largely distinct, and populated by different researchers, but they make a good case that there are many connections in the introduction, initial establishment, emergence and spread of the two types of invasives. Many of the same skills are used in the two fields, and in modeling the different steps involved, indicating that the disciplines can share skills and models. They build this around the so-called One Health model, which is reasonable, and they include some well thought out/diagrammatic models of the steps involved in analysis and control of each type of invasion/emergence. Overall a useful contribution to the fields involved, and it will be interesting to see if the cooperation between the fields can be accomplished.

RESPONSE: Thank you for these comments

A few issues to consider;

1) They seem to leave out or at least not explicitly include an important component of the One Health effort - the medical communities (human and veterinary).

Perhaps the invasive studies do not obviously connect up, but the emerging diseases would do?

RESPONSE: The reviewer is correct that One Health encompasses a very wide range of disciplines, and human and veterinary medical communities are important ones. This is now included (Page 3, Lines 64-65 and Page 4 Lines 102-103).

2) A few jargon words that I am not immediately sure of the definition - disservices? Maybe those can be defined or a glossary included.

RESPONSE: We have defined these in the text (Page 4 Lines 106-108) and checked that there are not other terms that go undefined in the text.

3) They do not explicitly seem to show how the fields are connected by invasive vectors (mosquitoes, ticks, others) and that provide a pathway for the diseases that take advantage of those to spread into new areas.

RESPONSE: We did not emphasise this as it has been the subject of previous reviews, although we have changed the text somewhat to indicate that this subject has been particularly identified as directly linking the disciplines (Page 3 Lines 82-84).

4) The term pre-adaptation is unclear to me, and perhaps not really an accurate definition of certain mutations. Not sure what is meant by "eco-evolutionary experience". Mutations arise and selected in an environment, which may favor spread in another environment - not sure if that is pre-adaptation as it implies some forshadowing of future selections in other environments?

RESPONSE: The terms pre-adaptation and eco-evolutionary experience have been widely used in invasion biology and for that reason we include them here. We hope now that these terms are better explained on Page 6 Lines 195-203.

5) Page 6 - ii) - Compatibility? For disease emergence this would also include population size, density and connectedness?

RESPONSE: Transmission of infectious diseases clearly depends on the characteristics of the pathogen (which we term in this article compatibility) and on the host population characteristics. The latter we include more specifically as the "environment" for this article (Page 7 Lines 230-231), but we also emphasise that while we have separated geography, compatibility and environment for convenience, the reality is that these are often mutually dependent in determining invasion/emergence (Page 7 Lines 251-254).

6) Page 6 - Would be "as for non-disease-causing invasive..."

RESPONSE: This has been corrected.

7) Page 8 - Not sure that EIDs are typically obligate parasites, or of vertebrates. Many could be opportunistic; also could equally well emerge in invertebrates (or plants or prokaryotes)?

RESPONSE: It is an interesting point to think in the wider context of emerging infectious diseases, and we have included mention of this in the Background (Page 3, Lines 57-62) and in the One Health section in response to reviewer 2 (Page 4 Lines 117-124). However, this article focuses on EIDs of public health significance (Page 3 Lines 56-57). It is true that some EIDs are not obligate parasites of vertebrates (e.g. Cryptococcus infection, Vibrios), but many EIDs of public health significance are, and this has driven an industry of modelling studies. To make this clear we have slightly re-worded this section (Page 8 Lines 285-286).

8) Page 9 - "relatively recently" is cited to a 2001 paper - almost a generation ago.

RESPONSE: We have changed this from "relatively recently" to "more recently", as the comparison is with the work in invasion biology which dates from the 1980s (Page 9 Line 332).

9) Page 9 - determining parameters of the different components of the emergence/invasion systems would be needed for proper control strategies?

RESPONSE: Yes that is right – thank you and now included (Page 9 Lines 341-343).

10) Page 10 - what is the data showing that funding is "orders of magnitude" less for invasion studies.

RESPONSE: Indeed this statement based on perception needs supporting, however obtaining the data is a long process so in the interest of timeliness we have changed the text here (Page 10, Lines 385-387).

11) Page 10 - why is only the European CDCD cited? Most counties have organizations with similar responsibilities that might be mobilized? What are the organizations that address invasive transmission?

RESPONSE: We have slightly changed this to identify a range of public health organisations in different countries, and to identify the range of organisations that are responsible for responding to invasive species (Page 11 Lines 416-427).

12) Figure 3 - This is an odd diagram and unclear - perhaps the barriers need to be extended across the different columns to make it clear what it is showing.

RESPONSE: We hope the modified version of this figure makes it more clear.

Reviewer: 2

Comments to the Author(s)

The paper you have submitted says, in the title, that it's a call for closer collaboration and, on p. 3 line 52, that it's a review. In p. 3 lines 46-7 you say the ideas are not new. In my view, your paper is a very useful and thorough survey of some relevant science. The science is embedded in phrases that say that both invasionists and epidemiologists 'require' (p 12 line 7) a joint One Health approach. You need to say, both in the title and early on, that the paper is both a call and a review and to connect the two more clearly.

RESPONSE: Thanks and good idea. One Health is now included in the title and that this is both a review and a "call" is included in the last paragraph of the background section.

In particular you need clearer arguments that invasion scientists and epidemiologists would produce better results than working separately and, also, why those two disciplines.

REPNSE: We feel that we have built an argument of the basis that i) the two fields have very similar skillsets and are working in parallel; and ii) have different ways of working that may be synergistic if put together. We have included what we hope is a more punchy final statement (Page 12 Lines 495-496).

Are not e.g. socio-economic factors important in the management of EIDs and invasions? Is there not an organisation and journal called EcoHealth working on that? See vol 15 part 2, June 2018 for a symposium on EIDs and economics.

RESPONSE: Yes economics is an important tool in developing policies on EIDs, although that does not mean that using economics as a tool is a One Health approach. There are many aspects of One Health involving interdisciplinary collaborations, and encouragement of integration of sciences is provided by journals such as EcoHealth. However here we are explicitly looking at the integration of invasion biology and EID epidemiology. We argue that invasions and EIDs are all part of One Health, but we are not trying to say that integration of invasion biology and EID epidemiology defines One Health.

Maybe you should also look at EPPO (The European and Mediterranean Plant Protection Organization) and give more attention to plant diseases.

RESPONSE: We had included the importance of invasions (including infectious diseases that do not affect humans) indirectly for human health although we now emphasise this in the revision (Page 4 Lines 117-124).

I think though that the chief weakness is that you don't say enough about One Health as an organisation. What sort of a body is it? What does it do? Has it had any results that have had an important effect on the management of human and /or animal diseases? Its web site gives the impression of a sleepy organisation just running meetings and not keeping its web site up to date.

RESPONSE: One Health is not an organisation but an approach to preventing and responding to infectious diseases and other health issues (<https://www.who.int/features/qa/one-health/en/>). There is a voluntary organisation called the One Health Initiative but this does not define One Health. We hope that the substantial revisions to the section on One Health makes this clearer (Page 4 Lines 99-105).

The arguments connecting One Health and a possible new UN (or other) organisation with your survey could be set out less vaguely, with better examples of how a One Health approach could help. Maybe you could be explicit about some situations where a One Health approach would clearly help and others where it would not be useful. For instance, could epidemiology help in managing pine invasions in the fynbos and, if so, how? And conversely, invasion science and managing AIDS in Cape Town?

RESPONSE: This is a good idea and we have now included three examples as suggested (Page 12 Lines 458-472).

The bulk of your paper is the survey of the similarities and differences between epidemiology and invasion science. Most of it is excellent and would be very useful to both communities. You put your over-view into a series of figures of a type that I personally don't find helpful but many people do. One Health is based on combing

the three fields called Human, Animal and Environmental in Fig 1, but the joint set of those three is empty in fig 1 (apart from an arrow passing through).

RESPONSE: The inclusion of the three fields as “circles” within the One Health concept (indicated by the green circle) is indeed confusing and the dashed circles have been removed.

In Fig 2 the boundary of Animal is oddly placed.

RESPONSE: This has been slightly changed but the “animal” circle explicitly surrounds those aspects (borehole drilling and changes in water consumption) that are affected by livestock production. The legend has been altered to emphasise this.

Fig 3 shows a scheme following one devised by Dr Richardson with barriers and there is an assertion that the first barrier to cross is a geographical one, as it was with WNV. But I would have thought that most EIDs have first to cross a species barrier, as HIV had to jump from chimpanzee to man or ebola from fruit bats (maybe) with no geographical barrier involved.

RESPONSE: We make the point that pathogens going from a wild animal to humans may be one of “compatibility” (i.e. the species barrier) that may require genetic changes that allow the species barrier to be crossed. However, in some cases, such as Ebola, there is no detectable inter-species barrier of compatibility, there is simply a barrier of humans coming into contact with infected wildlife, which is essentially a “geographic” barrier, albeit at a scale much smaller than generally considered by invasion biologists. We have emphasised this in the section on geographical barriers (Page 5 Lines 168-169). We have also clarified this in the legend of Fig 3.

Fig 4 gets as far as management planning with no management action, Which brings me to the most disappointing section, on p. 9 ‘B) Differences in risk management methods’ which is a classified reading list not a review and, in particular no results, no examples of real management. As the obvious difference between EIDs and classical biological invasions is the way they have to be managed, this is a major gap that should be filled. This section needs to be greatly expanded.

RESPONSE: We agree that the possibility for collaborations on management of invasions and EIDs is not covered in any detail in our article, but is worthy of much exploration. We made the point in the text of the original version that it is a subject of such a wide scope that it would require a dedicated review (Page 10 Lines 347-348). However, we have expanded the “Differences in risk management methods” section in response to this criticism (Page 10 Line 344 to Page 11 Line 427).

The other areas that I think are neglected are, as mentioned above, plant pathogens and, quite different, insect parasitoids whether introduced for biological control or, occasionally, introduced pests. And maybe other insect parasites such as leaf-miners.

RESPONSE: The subject of the article is explicitly EIDs of public health significance. However the reviewer is correct in inferring that techniques used by EID epidemiologists are highly applicable to studying plant pathogens, insect parasites

of plants and parasitoids and we now make this point in the text and add a reference (Page 8, Lines 272-274).

I would hope that such revisions would make a much more compelling case.

RESPONSE: We thank the reviewer for his/her insightful comments.

Appendix B

Dear Mr Dunn and Prof. Padian,

Many thanks to you and your reviewer for the comments on our article. We have made the minor changes you requested and hope these are suitable. Many thanks for helping us with this article and we look forward to seeing it published

Yours sincerely

Nick Ogden

Response to Reviewer 2 comments:

Thank you for your careful responses. I have just a couple of suggestions for wording. In lines 37 & 38 you say necessary and requires which have a touch of hype about them which may make people sceptical.

RESPONSE: These have been changed as follows:

"...approaches encompassing all three components are necessary respond to threats..." has been changed to "...approaches encompassing all three components are needed to respond to threats..."

"We argue that sustainable development requires explicit consideration of biological invasions within One Health." Has been changed to "We argue that for sustainable development biological invasions should be explicitly considered within One Health."

In lines 103, 881 you say actor which, in English English, is almost always someone on a stage or in a film. Your usage may be commoner in other English speaking countries but for a London based publication I would suggest participant or agent (or even colleague or collaborator).

RESPONSE: These have been changed as follows:

For the first (in the "One Health concept" section), "actors" has been changed to "collaborators."

For the second (in the legend of Fig 4), "The rows of boxes represent the different actors in the fields of EIDs and biological invasions" has been changed to "The rows of boxes represent the different fields of involved in responding EIDs and biological invasions."